# Mesenchymal Stem Cell-Derived Extracellular Vesicles for Bone Defect Repair

**DOI:** 10.3390/membranes12070716

**Published:** 2022-07-19

**Authors:** Dongxue Wang, Hong Cao, Weizhong Hua, Lu Gao, Yu Yuan, Xuchang Zhou, Zhipeng Zeng

**Affiliations:** 1School of Sport Medicine and Rehabilitation, Beijing Sport University, Beijing 100084, China; wdx17861520156@163.com (D.W.); 18811393829@163.com (W.H.); gaolukf@163.com (L.G.); 2School of Kinesiology, Shanghai University of Sport, Shanghai 200438, China; chhcaohong@163.com (H.C.); yuanyumail@126.com (Y.Y.)

**Keywords:** bone graft, exosomes, extracellular vesicles, mesenchymal stem cells, tissue engineering

## Abstract

The repair of critical bone defects is a hotspot of orthopedic research. With the development of bone tissue engineering (BTE), there is increasing evidence showing that the combined application of extracellular vesicles (EVs) derived from mesenchymal stem cells (MSCs) (MSC-EVs), especially exosomes, with hydrogels, scaffolds, and other bioactive materials has made great progress, exhibiting a good potential for bone regeneration. Recent studies have found that miRNAs, proteins, and other cargo loaded in EVs are key factors in promoting osteogenesis and angiogenesis. In BTE, the expression profile of the intrinsic cargo of EVs can be changed by modifying the gene expression of MSCs to obtain EVs with enhanced osteogenic activity and ultimately enhance the osteoinductive ability of bone graft materials. However, the current research on MSC-EVs for repairing bone defects is still in its infancy, and the underlying mechanism remains unclear. Therefore, in this review, the effect of bioactive materials such as hydrogels and scaffolds combined with MSC-EVs in repairing bone defects is summarized, and the mechanism of MSC-EVs promoting bone defect repair by delivering active molecules such as internal miRNAs is further elucidated, which provides a theoretical basis and reference for the clinical application of MSC-EVs in repairing bone defects.

## 1. Introduction

A critical-size bone defect is defined as the smallest size intra-osseous wound in a particular bone and species that will not heal spontaneously during the lifetime of the animal [1,2]. Critical-size bone defects are often caused by trauma, bone tumors, and infection [3]. At present, the clinical strategies for repairing bone defects mainly include autograft, allograft, and synthetic biomaterial graft [4,5]. Autologous bone transplantation is generally considered to be the gold standard for repairing bone defects, although there are many limitations, including the high morbidity of donor bone, limited availability, and unpredictable autologous absorption [6,7]. The clinical application of pure bone allograft is limited due to its susceptibility to immune rejection [8]. In 1993, Langer et al. [9] first proposed the potential strategy of applying tissue engineering to regenerative medicine. Bone tissue engineering (BTE) is mainly used for large-scale bone defect repair, which includes three basic components: bioscaffolds, seed cells, and osteoinductive factors [10]. In recent years, BTE has made great progress in the treatment of bone regeneration. Bioactive materials such as scaffolds combined with mesenchymal stem cells (MSCs) and their secreted factors have been widely used in the repair of bone defects [11]. MSCs are a heterogeneous mesenchymal stem cell subpopulation, including bone marrow mesenchymal stem cells (BMSCs) [12], adipose mesenchymal stem cells (ADSCs) [13], umbilical cord mesenchymal stem cells (UCMSCs) [14], and induced pluripotent stem cells (iPSCs) [15], with multi-directional differentiation potentials such as osteoblasts, chondrocytes, and adipocytes [16]. Therefore, stem cell therapy is considered a potential strategy for bone defect regeneration [17,18]. However, there are still some disadvantages, such as phenotypic changes, low homing efficiency, and low cell viability, which limit the application of MSCs in bone defect repair [13]. In recent years, cell-free therapy for bone regeneration has attracted extensive attention [19,20]. Accumulating evidence indicates that the positive effect of MSCs on tissue repair is to stimulate the activity of tissue-resident receptor cells through paracrine rather than directly differentiate into parenchymal cells to repair or replace damaged tissue [21,22]. Among the many cytokines secreted by MSCs, extracellular vesicles (EVs) are nano-phospholipid bilayer vesicles that can be secreted by almost all cells and are formed by “budding” directly from the plasma membrane. According to their diameter, composition, and origin, EVs can be divided into exosomes (Exos), microvesicles, and apoptotic bodies [23]. Among them, Exos are the most attractive category of EVs. Exos are bilayer lipid-membrane vesicles with a diameter between 40 and 150 nm, which carry a rich variety of nucleic acids, proteins, miRNAs, mRNAs, and lipid substances. As a nanocarrier for cell-to-cell communication, Exos are released extracellularly to exert regulatory effects by transferring intrinsic cargo to target cells via ligand-receptor interactions, endocytosis, or direct membrane fusion [24]. Numerous studies have shown that Exos obtained from MSCs are considered to be the main mediator of the therapeutic effect of MSCs without the limitations of MSC therapy [16,25,26]. Recently, in bone regenerative medicine, the research of MSC-derived Exos (MSC-Exos) has become an attractive hotspot [27]. Many studies have shown that bioactive materials such as hydrogels or scaffolds loaded with MSC-derived EVs (MSC-EVs) can significantly promote osteogenesis and angiogenesis, thereby effectively repairing bone defects [28,29].

However, the research on MSC-EVs promoting bone defect repair is still in the preliminary stage, and the specific mechanism is still unclear. In this paper, the application progress of MSC-EVs combined with hydrogels and/or scaffolds in repairing bone defects and the potential mechanism of MSC-EVs promoting osteogenesis or angiogenesis are comprehensively summarized. This review provides the latest evidence and perspectives on the effective application, potential risks and solutions of MSC-EVs combined with hydrogels and/or scaffolds and the potential mechanism of MSC-EVs in enhancing bone regeneration, which will provide a theoretical basis for further application of MSC-EVs to promote bone defect repair.

## 2. Materials and Methods

This paper aims to summarize the advanced application of bioactive materials combined with MSC-EVs utilized in repairing bone defects and the mechanism of MSC-EVs promoting bone regeneration. A literature search was performed in electronic databases, including PubMed, Medline, OVID, and Web of Science, by using the following keywords: “tissue engineering”, “bone defects”, “EVs”, “extracellular vesicles”, “Exos”, “mesenchymal stem cells” and “exosomes”. Documents published in English were selected, and the articles were further screened to identify their relevance to this review (as shown in Figure 1). This narrative review briefly discusses the application of bioactive materials combined with MSC-EVs in repairing bone defects and the underlying mechanism of MSC-EVs to promote bone regeneration, as well as the limits of current studies.

## 3. Application of MSC-EVs in Bone Defects

### 3.1. Application of EVs Parent Cells

As key mediators of cell-to-cell communication, Exos play important roles in a variety of biological roles and injury repair processes, such as regulating immune response [30], promoting the repair of traumatic brain injury [31], and stimulating bone tissue regeneration [32]. Studies have shown that in the treatment of bone tissue regeneration, MSC-Exos stimulate osteoblast proliferation and angiogenesis and inhibit osteoclast maturation by delivering endogenous cargo [33]. Since Exos generally have similar therapeutic properties to their parent cells, Exos derived from different MSCs have different effects on bone regeneration and repair [34,35]. Therefore, in the strategy of applying MSC-Exos to bone regeneration, it is important to select an appropriate source of parental cell types [36].

The sources of parental MSCs of Exos mainly include BMSCs [12], ADSCs [13], UCMSCs [14], iPSCs [15], etc. As shown in Table 1, among these cells, BMSCs are the most commonly used cell source for tissue engineering [37]. Numerous studies have shown that BMSC-derived Exos (BMSC-Exos) can stimulate the proliferation and osteogenic differentiation of BMSCs, thereby promoting osteogenesis, angiogenesis, and bone mineralization in critical bone defect models [38]. Compared with BMSCs, human adipose mesenchymal stem cells (hADSCs), mainly derived from adipose tissue, are easier to obtain. hADSCs are widely distributed in the human body and proliferate rapidly, which is an ideal type of MSC for improving Exos production [39]. However, compared with BMSCs, hADSCs may have less osteogenic differentiation potential [40]. Some scholars compared the osteogenic differentiation potential of rat BMSC-Exos and ADSC-derived Exos (ADSC-Exos) in osteogenesis-induced (OI) conditions. The results showed that BMSC-OI-Exos significantly up-regulated the expression of osteogenesis-related genes and proteins, including Col I, Runx2, BSP, OCN, BMPR-IA, and BMPR-II, while ADSC-OI-Exos only up-regulated BSP, BMPR-IA and, BMPR-II, which suggested that ADSC-OI-Exos may have a lower osteoinductive ability compared with BMSC-OI-Exos [41]. In addition, hADSC-derived Exos (hADSC-Exos) have shown great therapeutic potential in the treatment of ischemic diseases, indicating that hADSC-Exos may also play a potential role in bone regeneration by promoting angiogenesis [42]. Human umbilical cord mesenchymal stem cells (hUCMSCs) have also attracted much attention in BTE. hUCMSCs are a primitive population of MSCs obtained from Wharton’s jelly, the main component of the human umbilical cord matrix [43]. hUCMSCs can indirectly promote bone regeneration by inducing angiogenesis rather than directly promoting osteogenic or chondrogenic differentiation. Compared with other MSCs derived from bone marrow or fat, hUCMSCs showed higher pluripotency [44,45] and stronger pro-angiogenesis properties [46]. In addition, hUCMSCs are derived from human postnatal waste tissues and were abundant in source, which is a rich source without ethical and moral controversies, showing greater clinical potential [47]. Similar to hUCMSCs, iPSCs also have no immune rejection or ethical issues and can proliferate indefinitely; however, iPSCs may have tumorigenic risks [48]. In recent years, studies have shown that MSCs derived from iPSCs (iPS-MSCs) have the advantages of both iPSCs and MSCs. iPS-MSCs can maintain their self-renewal ability after 40 passages. Importantly, iPS-MSCs are not tumorigenic [49]. In the treatment of bone defects, iPS-MSCs show strong proliferative and immunomodulatory abilities to promote bone regeneration [15]. Exos derived from iPS-MSCs also have similar properties to their parental cells. Therefore, iPS-MSCs are an excellent cell source for Exos to promote bone regeneration [22]. In addition to the above-mentioned MSCs, some local tissue-derived MSC-Exos also have good application potential. Stem cells from human exfoliated deciduous teeth (SHEDs), which are immature MSCs with multiple differentiation potential, can be easily obtained in a non-invasive way without ethical problems [50]. Compared with BMSCs, SHEDs have a stronger proliferative capacity due to the abundant secretion of growth factors such as fibroblast growth factor 2 (FGF2) and transforming growth factor-β2 (TGF-β2) [51]. Moreover, compared with pulp stem cells, SHEDs have a stronger capacity to promote bone mineralization [52]. SHED-derived Exos, in combination with tricalcium phosphate (β-TCP), have been shown to enhance alveolar bone regeneration by promoting angiogenesis and osteogenesis [53]. In addition to the source of parental MSCs, the therapeutic effect of Exos may also be influenced by some underlying diseases of the donor. A recent study showed that BMSC-Exos obtained from the T_1_DM rat model was significantly less effective in promoting BMSC osteogenic differentiation and endothelial cell angiogenesis than BMSC-Exos obtained from normal rats [35]. This suggests that Exos from donors with chronic underlying diseases have a poor therapeutic effect, or even the opposite therapeutic effect, which may not be suitable for regenerative medicine [54]. Further research is needed to explore the potential therapeutic effects of donor-derived Exos in disease states.

### 3.2. Application of MSC-EVs in BTE

The regeneration of critical bone defects caused by bone tumors, injuries, and other bone diseases requires bone grafting to replace the defective bone. EV-mediated BTE therapy has become an attractive bone defect replacement strategy in recent years [55]. Accumulating studies have shown that EV-loaded hydrogels or scaffolds, alone or in combination, can significantly promote local bone regeneration (as shown in Figure 2).

#### 3.2.1. Application of MSC-EVs Combined with Hydrogels

Hydrogels are physically or chemically cross-linked three-dimensional (3D) hydrophilic polymer networks that can adsorb large amounts of water without being dissolved [56]. Biomedical hydrogels are structurally similar to the natural extracellular matrix, which have attracted extensive attention in regenerative medicine due to their properties of tissue-like water content, good biocompatibility, and easy implantation. In regenerative medicine, hydrogels, as an effective carrier, are widely used in drug delivery and active molecule encapsulation matrix for tissue repair, including bone tissue repair [57]. In bone tissue regeneration, biopolymer hydrogels can be injected directly into bone defects and flow over complex and irregular bone surfaces, matching the gaps between implants and randomly shaped defects and increasing the contact area [58]. In addition, the porous structure of hydrogels can continuously release bioactive factors, including EVs/Exos, which is beneficial for repairing bone defects in BTE [58]. Studies have shown that Exos coated in biodegradable hydrogels can still exhibit the desired therapeutic effect [59,60]. Hydrogels provide a 3D matrix for Exos that prevents the dispersion of Exos and maintains its local concentration, which enables controlled release of Exos for sustained efficacy. Hydrogel-released Exos at bone defect sites can reduce Exos consumption and ectopic effects to promote angiogenesis and osteogenesis [61]. Some scholars found gelatin nanoparticle hydrogels combined with hADSC-Exos can accurately transport the hADSC-Exos to the target site and effectively promote bone healing [62]. In addition, the hydrogel 3D microenvironment can also enhance Exos activity and affect the interaction of EV integrin membrane protein between cells and the cell matrix to promote the proliferation and differentiation of osteoblasts and bone angiogenesis in a bone regeneration environment. Yu et al. [4] demonstrated that Exos derived from periodontal ligament stem cells (PDLSCs) encapsulated in a hydrogel 3D microenvironment exhibited enhanced osteoinductive ability and could significantly promote bone defect repair in rats. Another study demonstrated alginate hydrogels combined with EVs showed increased interactions with other cells, cell aggregation, and prolonged long-term viability, which in turn promoted osteogenesis [61]. Therefore, a variety of special properties of hydrogels can be combined with MSC-EVs to effectively treat bone defects.

At present, the commonly used biomedical hydrogels mainly include natural and synthetic polymer hydrogels, such as gelatin, hyaluronic acid hydrogel (HA-gel), chitosan, polyethylene glycol (PEG), etc. [63]. Compared to natural materials, synthetic polymers have basic structural units, well-defined properties, including porosity and degradation time, and known mechanical properties, making synthetic polymer hydrogels more suitable for specific bone tissue regeneration applications [64]. In addition, synthetic polymers are suitable delivery vehicles for EVs in bone defect repair due to their reliable source of material, long shelf life, and low risk of immunogenicity [65]. However, due to their fragile network structure, the mechanical properties, stability, and cell adhesion of traditional hydrogels are usually weak, which hinders their application as bone structural materials [66]. To overcome this limitation, hydrogels are often modified by chemical, physical, and biological methods to prepare high-performance hydrogels with enhanced mechanical properties, stability, cell adhesion, and other properties [63]. Ma et al. [67] used propionic acid (CA) with a catechol group to modify an injectable thermosensitive hydrogel through a covalent bond. The cross-linking network formed by the interaction between the catechol group of CA and the collagen molecules in hydrogels can effectively improve the mechanical properties of hydrogel [68]. In addition, the combination of two hydrogels to prepare new hydrogels is also a common way of hydrogel modification. Previous studies revealed that the combination of a gelatin derivative, gelatin methacrylate, and a PEG derivative, polyethylene glycol diacrylate, can prepare a hydrogel with good cell adhesion and stronger mechanical properties [69]. Hydrogels with good performance can be obtained by various methods; however, how to effectively load bioactive molecules such as EVs/Exos into hydrogels and improve their long-term retention at defect sites is also a challenge in BTE. Currently, the commonly used method is to directly mix Exos with hydrogels [70], which may lead to low loading efficiency, structural destruction of Exos, and hindered osteogenic potential [71]. Recent studies have demonstrated that substances such as peptides can be used to effectively fix EVs/Exos, showing excellent drug delivery ability, which can effectively deliver EVs/Exos [72]. Huang et al. [29] found that integrin membrane proteins in MSC-EVs could combine with collagen and fibronectin-derived peptides secreted by cells. Subsequently, the adhesive peptide (RGD) was added to alginate hydrogels containing EVs overexpressing bone morphogenetic protein 2 (BMP-2) to target cellular integrins. The results showed that EVs could be effectively fixed to hydrogels through the interaction of protein domains on the cell membrane [29]. In another study, some scholars incorporated Exos into an injectable thermosensitive hydrogel by constructing fusion peptides, which also enhanced the retention of Exos and improved the biological activity of Exos [67]. The above results indicated that the addition of bioactive substances such as peptides can enhance the fixation and retention of EVs without impairing the structural integrity and biological activity of EVs. This enhanced fixation and retention may prolong the presence of EVs at healing sites, leading to increased efficiency of delivery as well as functionality. It should be noted that the mechanical properties, gelation time, biocompatibility, and biological functions of hydrogels vary due to different raw materials and synthesis conditions in the practical application of hydrogels [73]. Some studies suggest that chitosan hydrogel has good biocompatibility, hemostasis, and chemical activity [74], while alginate is a biologically neutral material with adjustable mechanical properties and biodegradation [75]. In addition, since the efficacy of MSC-EVs encapsulated in hydrogels on bone regeneration largely depends on the design and function of hydrogels, attention should be paid to the different characteristics of hydrogels when they are used as delivery materials [73]. Zhang et al. [76] prepared an injectable thermosensitive chitosan hydrogel with good biodegradability and biocompatibility, which can improve the stability and retention of human placental MSC-Exos, and effectively enhance bone regeneration by promoting angiogenesis.

In summary, due to its special structure, hydrogels can effectively fill the bone defect area. When combined with EVs and other bioactive molecules, the porous hydrogels can realize the sustained release of EVs. In addition, on the one hand, effective encapsulation of hydrogels can maintain the concentration of EVs and provide a 3D microenvironment similar to the natural extracellular matrix, enhancing the activity of EVs and promoting injury repair; On the other hand, it reduces the possibility of contact with other tissues during transportation, which improves the accuracy and effectiveness of transportation, thereby maximizing the osteogenic effect of EVs. Traditional hydrogels have shortcomings such as insufficient mechanical properties, stability, and adhesion. Recent studies have found that hydrogels can be modified by adding active substances such as peptide sequences to improve their performance and achieve long-term retention of EVs. In addition, different types of hydrogels should be carefully selected to effectively exert their different properties in BTE. However, when applied to critical bone defects with large distances or poor biomechanical environment, it has to be admitted that the poor mechanical properties of hydrogels still cannot satisfy the effective repair of the damage, which makes the development and application of scaffold materials attract extensive attention in BTE.

#### 3.2.2. Application of MSC-EVs Combined with Scaffolds

In the repair of bone defects, bioactive scaffolds are often used as bridging materials to promote bone regeneration [9]. BTE scaffolds are not only bioabsorbable and biodegradable but also can provide temporary mechanical support for the bone at the implant site, with good mechanical properties. In addition, scaffolds have a highly porous and 3D structure that ideally mimics the porosity, pore size, and interconnectedness of native bone. This specific structure promotes cell attachment and proliferation, providing space for the growth and vascularization of new tissue [77,78]. Palma et al. [79] reported the influence of different formulations of bone grafts in providing an adequate scaffold, thus emphasizing the importance of the type of carrier in the three-dimensional distribution of particles and also space provision in new bone formation. The results showed that the lyophilized form carrier created a more homogenous interparticle spacing, allowed a more suitable particle distribution and stabilization, and provided a required space which is crucial for proper cellular and vascular colonization, then promoting a faster bone regeneration with relevant clinical benefits [79]. Similarly, compared with compacted materials, a biocompatible 3D porous scaffold could ensure a uniform spacing and stable distribution of MSC-EVs [80]. The pores or the space provision of scaffolds can assure an adequate environment for growth factors and nutrients, realizing the constant flow of nutrients, cells, and growth factors from the outer portion to the core of the scaffold and promoting bone regeneration [81,82]. In addition, scaffold surface porosity and the related micro- and nano-morphology directly influence cell behavior, stimulating proper communication among the resident cells [80,83,84]. There is no consensus concerning the more appropriate porosity value or pore size currently. However, when mechanical properties are satisfied, over 90% of studies recommended high porosity values, with a wide range of pore sizes from 10 to at least 200 µm [83,85]. In BTE, bioscaffolds with good mechanical properties provide attachment for osteoblasts, which proliferate and differentiate into osteocytes. Subsequently, the implanted scaffold is gradually degraded and replaced by mature osteocytes, and finally, the typical bone structure is restored [86]. In addition, specific surfaces of some osteogenesis-induced scaffolds support osteoblastic cell differentiation and the expression of the osteoblastic phenotype [87]. Studies have shown that the classical β-TCP porous scaffolds have good biocompatibility and are widely used in the clinical application of bone repair and regeneration [88]. However, various scaffolds, including β-TCP scaffolds, have limited repair capacity for critical bone defects due to a lack of osteoinductive activity [89,90]. Only relying on the scaffold itself as a bone graft material cannot achieve satisfactory bone defect repair [77,91]. Accumulated studies have proved that MSC-EVs can effectively enhance the osteoinductive performance of scaffold materials as bioactive molecules. Therefore, as a bioactive material for carrying MSC-EVs, the combined application of bioscaffolds and MSC-EVs has a better regenerative effect in repairing bone defects [22]. Zhang et al. [22] demonstrated for the first time that human iPS-MSC-derived Exos (hiPS-MSC-Exos) can be used in combination with β-TCP to significantly promote osteogenesis in the rat calvarial defect model. Exos loaded on hiPS-MSC-Exos/β-TCP scaffolds are internalized into human bone mesenchymal stem cells (hBMSCs), which can stimulate the proliferation and differentiation of endogenous BMSCs and promote the recruitment of BMSCs to specific sites in bone defects to enhance new bone formation, indicating that Exos can enhance the osteogenic activity of β-TCP, and the potential mechanism of Exos promoting bone regeneration may be the activation of endogenous BMSCs at the bone defect sites. In addition, Xin et al. [92] demonstrated that hiPS-MSC-Exos/β-TCP combined with scaffolds can effectively promote neovascularization in the osteoporosis rat calvarial defect model, and the area of newly formed vessels increases with the increase of hiPSC-MSC-Exos concentration. Furthermore, MSC-Exos from gums or fat in combination with polymer or polymer/calcium silicate composite scaffolds also enhanced bone regeneration [93,94]. All the above results confirm the potential of MSC-EVs combined with scaffolds in repairing bone defects.

There are various types of scaffolds in BTE, including collagen sponge scaffolds [95], β-TCP scaffolds [22], hydroxyapatite (HA) scaffolds [14], calcium sulfate cement scaffolds [96], bioactive glass [41], polycaprolactone (PCL) scaffolds [97], and other innovative synthetic scaffolds [19], which should be selected appropriately according to their different properties when applied to the regeneration of bone defects. For example, compared with other bone substitutes, the translucent collagen sponge is an ideal scaffold for the detection of MSC-Exos in periodontal tissue regeneration and is commonly used as a scaffold material for carrying Exos in periodontal defect models [98]. In addition, native EVs have poor osteoinductive ability. The secretion of EVs and the expression profiles of their intrinsic cargo are influenced by different cellular microenvironments. Different delivery materials or culture conditions can modulate the microenvironment of parental cells and alter the abundance of EV cargo by giving appropriate physical, chemical, and mechanical stimuli, thereby affecting the function of EVs [99,100,101]. Therefore, currently, parental MSCs are often pretreated with hypoxia, chemical agents, or cytokines to obtain EVs with enhanced osteoinductive properties [102,103]. The functional engineered EVs prepared by various methods combined with different scaffold materials can effectively promote the repair of bone defects [104,105]. Liang et al. [106] found that angiogenesis was significantly promoted when dimethyloxalylglycine (DMOG)-pretreated MSC-Exos loaded by classical HA scaffolds in a rat calvarial defect model. DMOG is a small angiogenic molecule that regulates the stability of hypoxia-inducible factor-1α (HIF-1α) by inhibiting proline hydroxylase and can simulate cell hypoxia at normal oxygen levels [107,108]. However, hypoxia can improve the regulatory ability of MSCs to induce angiogenesis, osteogenic differentiation, and anti-apoptosis by activating the expression of HIF-1α, thereby changing the expression profile of MSC-Exo content [109]. Furthermore, preparing genetically modified hBMSCs by constitutively expressing osteogenic-related genes or proteins is also a common strategy to improve the osteoinductive properties of EVs. The functionally engineered EVs prepared by various methods combined with different scaffold materials can effectively promote the repair of bone defects [105,106]. Huang et al. [110] generated functionally engineered EVs with enhanced osteoinductive properties by constitutively expressing BMP-2 in hBMSCs, which showed a good effect on bone defect repair. Moreover, Ying et al. [111] loaded BMSC-Exos carrying mutant HIF-1α (BMSC-Exo-HIF-1α) into β-TCP scaffolds and found that the effect of BMSC-Exo-HIF-1α combined with β-TCP scaffolds in promoting angiogenesis and new bone regeneration was significantly better than that of BMSC-Exos combined with β-TCP scaffolds in the rat calvarial defect model. Although EVs with enhanced osteogenic capacity obtained by various means show good regenerative therapeutic effects, to date, most of the bioactive molecules loaded by carrier systems, including EVs, show explosive release, which may stimulate early bone resorption, leading to brittle bone formation or reduced bone formation, thereby impairing the osteogenic properties of EVs [112]. Therefore, to achieve the slow and sustained release of EVs required for treatment and improve the efficiency of drug administration, many explorations have been carried out. Through the synthesis of degradable copolymer poly(lactic-co-glycolic acid) (PLGA) and metal-organic frameworks (MOFs), Yue et al. [19] developed an Exo-functionalized PLGA/Mg-GA MOF (PLGA/Exo-MG-GA MOF) scaffold with a unique nanostructure that achieved the controlled release of cargo, thereby effectively promoting osteogenesis and angiogenesis and alleviating inflammation. In addition, some scholars have explored the application effect of the combination of synthetic scaffolds and reagents. Qayoom et al. [96] also achieved the controlled release of Exos with long-term effects by using independently synthesized calcium sulfate/nano-hydroxyapatite nano-cement (NC) as the carrier, combined with BMP and zoledronate (ZA), an anti-bone absorption agent. This is attributed to the bionic properties and interaction potential of the NC matrix with BMP and ZA. Previous studies have shown that the explosive release of Exos may be caused by direct adsorption of Exos on the scaffold due to the way of solution infusion. Some scholars believe that the explosive release of Exos may be controlled by surface chemical modification of the scaffold [13,92]. Lin et al. [113] designed a hierarchical mesoporous bioactive glass (MBG) scaffold, which was inherently osteoinductive and could provide structural bioactivity maintenance, which was applied as a carrier for lyophilized Exos in its micron-sized porosities. Further research found that the controlled release of BMSC-Exos and the maintenance of the biological activity were achieved by lyophilized delivery of BMSC-Exos onto graded MBG scaffolds through the sheltering of micropores on the surface of the scaffolds [41]. Recent studies have shown that active lyophilized Exos obtained by adding lyoprotectants may enable more stable and lower-cost long-term preservation of Exos during delivery, although this research has not been widely recognized [114]. The above results demonstrate that controllable release and even long-term preservation of EVs can be achieved by rationally modified scaffolds, thereby effectively improving the bone regeneration effect of EVs.

In summary, loading EVs onto various bioactive scaffold materials is a common strategy for bone defect repair in BTE. However, the osteogenic activity of native EVs may not be sufficient for critical bone defect repair. Engineered EVs, such as pre-treatment of MSCs by genetic engineering, combined with synthetic scaffolds can significantly improve the osteoinductive ability and promote angiogenesis and osteogenesis in bone defect models. In addition, the development of innovative scaffold materials or the application of lyophilized Exos may be a new strategy to achieve effective EV delivery and controlled release to maximize the osteogenic advantages of EVs in BTE. However, it has to be admitted that there are some problems in loading EVs with scaffolds alone: (1) it is still difficult to achieve precise sustained release; (2) the surface environment of the scaffolds may affect the osteogenic properties of EVs; (3) the filling effect and biocompatibility of scaffolds on bone defects are also inferior to hydrogels.

#### 3.2.3. Application of EVs Combined with Hydrogels and Scaffolds

Bioscaffolds with good mechanical and stable properties can fill critical bone defects, while hydrogels with good hydrophilicity can effectively fill complex and irregular bone defects due to their gel-like properties. In addition, hydrogels with good biocompatibility and encapsulation properties have better performance in sustained-release EVs compared to bioscaffolds [56,77,78,115]. Therefore, the combination of hydrogels and scaffolds can not only provide good mechanical stability but also effectively fill bone defects based on irregular defect shapes and encapsulate EVs for controlled release. Currently, the application of EVs loaded by composite bioscaffolds combined with hydrogels is very attractive in bone regenerative medicine [116]. A recent study showed that the incorporation of a novel injectable hydrogel, polyethylene glycol maleate citrate (PEGMC), into β-TCP scaffolds effectively enhanced its composite strength and osteoinductive properties [117]. Subsequently, EVs were loaded into the composite PG/TCP (PEGMC + β-TCP) for application in bone defect repair. It was found that the novel composites loaded with EVs improved the microenvironment of BMSCs and induced high-rate and high-quality bone regeneration by promoting angiogenic activity [117]. Similarly, in another study, the combined use of HA-Gel hydrogel with a customized nano-hydroxyapatite/poly-ε -caprolactone (nHP) scaffold capable of matching the shape and size of the bone defects also significantly enhanced the angiogenic activity of UCMSC-Exos, demonstrating a strong ability to promote bone formation by inducing angiogenesis [14]. In this study, both above-mentioned composite scaffolds showed good biocompatibility. In addition, the customized nHP scaffold showed excellent mechanical support properties, while the HA-Gel hydrogel filled the pore structure of the scaffold and achieved the long-term sustained release of UCMSC-Exos. Moreover, to achieve the sustained release of Exos without compromising its biological activity, Swanson et al. [118] developed a PLGA-PEG-PLGA hydrogel microsphere as an Exos delivery system. Further, Exo-loaded PLGA-PEG-PLGA hydrogel microspheres were filled on poly (L-lactic acid) (PLLA) scaffolds, and it was found that the activity of Exos was not impaired and the repair of rat calvarial defects was significantly promoted. In addition, some studies have found that the new composite material obtained by the combination of solid scaffold materials and hydrogels not only has better mechanical properties but also shows some other unique properties such as self-healing [119]. Self-healing materials can repair themselves in a short time after being damaged, which can prevent the inhibitory effect of soft tissue infiltration on bone regeneration, thereby ultimately promoting bone damage repair [120]. Li et al. [47] successfully prepared a self-healing material (CHA/SF/GCS/DF-PEG) with good mechanical properties and plasticity by using CHA as the main component of the bone graft material, combined with silk fibroin (SF), glycol chitosan (GCS) and difunctionalized polyethylene glycol (DF-PEG). The CHA/SF/GCS/DF-PEG composite material is a promising scaffold for Exos because of its ideal structure and physical properties. The combined application of UCMSC-Exos with this composite material can effectively repair bone defects in rats by increasing BMP-2 and collagen deposition and promoting angiogenesis [47]. In addition, Liu et al. [115] prepared a composite hydrogel system with good adhesion and antibacterial properties modified by zeolitic imidazolate framework-8 (ZIF-8) nanoparticles, which can enhance the stability of the implanting environment after bone transplantation and promote bone repair. As a key member of MOFs, nanoscale ZIF-8, which sustainably releases Zn^2+^ and plays an active role in osteogenesis, angiogenesis, and antibacterial processes, is an effective modified material in BTE [121,122].

In summary, the combined application of hydrogels and scaffolds can obtain a composite bone graft material with good mechanical properties, biocompatibility, and encapsulation ability, which effectively makes up for the deficiency of hydrogels or scaffolds alone (As shown in Table 2). The combined application of the two materials, on the one hand, can provide the mechanical properties required for bone defect repair and fill critical bone defects with high stability. On the other hand, it can enhance the activity of EVs and make it possible to realize efficient delivery, long-term preservation, and sustained release of EVs, which is also of great significance for the effective repair of bone defects. In addition, the new graft material obtained by combining the hydrogels with the scaffolds also exhibits good self-healing, adhesion, and antibacterial properties, as well as other properties that are beneficial to the repair of bone defects. However, at present, the research on the application of complex composite materials in bone defect repair is insufficient due to the variety and complexity of raw materials. It should be noted that the above-mentioned new composite materials cannot be directly used for the treatment of clinical bone defects due to the high cost and complicated preparation process. Further research is needed to develop a low-cost and high-efficiency composite delivery system to deliver EVs in BTE.

## 4. Mechanism of MSC-EVs in Repairing Bone Defects

Accumulated studies have shown that MSC-EVs can induce osteogenesis and angiogenesis, regulate immune activity, and inhibit osteoclast activity, ultimately promoting the repair of bone defects by directly transferring their internal cargo and subsequent regulating downstream signaling cascades in target cells [123,124]. It has been reported that MSC-EVs contain a variety of biologically active molecules such as nucleic acids, proteins, and lipids. Among them, miRNAs are one of the most attractive intrinsic cargo of Evs in recent years [125]. miRNAs are small endogenous non-coding RNA, which are one of the main functional components of Exos. Numerous studies have reported that miRNAs play a key role in cell communication and are widely involved in bone metabolism regulation through post-transcriptional modification [126,127,128]. Previous studies have revealed that miRNAs contained in Exos are the main regulators that promote osteogenic differentiation [129]. Wang et al. [130] showed that the ability of T_2_DM rat-derived BMSC-Exos to promote osteogenic differentiation was significantly impaired, while up-regulating the level of miR-140-3p in Exos could restore its normal osteogenic ability. Further studies revealed that plxnb1 is a direct downstream mRNA target of miR-140-3p, which is closely related to bone formation. Therefore, miR-140-3p may promote bone formation by inhibiting the plexinB1/RhoA/ROCK signaling pathway. In addition, Chen et al. [131] found that hADSC-Exos overexpressing miR-375 could significantly enhance osteogenic differentiation. miR-375, as a positive regulator of osteogenic differentiation of BMSCs, can inhibit insulin-like growth factor binding protein 3 expression and subsequently activate insulin-like growth factor activity, ultimately promoting osteogenic differentiation [132]. Another study found that miR-1260a in Exos was up-regulated after pretreatment with Fe_3_O_4_ magnetic nanoparticles and a static magnetic field, which not only promoted osteogenic differentiation but also effectively induced angiogenesis [11]. Previous studies have confirmed that the up-regulation of miR-1260a directly inhibits HDAC7 and COL4A2 expression, while HDAC7 [133] and COL4A2 [134] can inhibit Runx2 activity and angiogenesis, respectively. Therefore, miR-1260a plays a key role in promoting osteogenic differentiation and angiogenesis by targeting HDAC7 and COL4A2. Similarly, another study found that the increased expression of miR-21 within UCMSC-Exos was associated with enhancing angiogenesis in repairing rat cranial defects. Further investigation showed that miR-21 can inhibit NOTCH1, a key pathway for regulating angiogenesis, and then induce the expression of VEGFA and HIF-1α to promote angiogenesis [14,135]. In addition to promoting osteogenesis and angiogenesis, other studies have found that Exo-derived miRNAs can also participate in immune regulation and inhibit osteoclast activity in the process of bone defect repair [62]. However, due to the large number of miRNAs encapsulated in EVs and the possible interaction between different miRNAs to form complex network regulation, the underlying mechanism is still not fully understood. Moreover, in addition to miRNAs, EVs also contain a large number of proteins, lipids, and other types of nucleic acids, such as long non-coding RNAs, circular RNAs, and mRNAs. Therefore, not only the regulatory mechanism of EV-derived miRNAs in bone defect repair needs to be further elucidated, but the role of other intrinsic molecules of EVs also needs to be explored.

### 4.1. MSC-EVs Repair Bone Defects by Promoting Osteogenic Differentiation

Enhancing the activity of osteoblasts and inhibiting the activity of osteoclasts are the key factors in bone defect repair. Previous studies have shown that MSC-EVs promote the recruitment of endogenous MSCs to bone defect sites, thereby activating and promoting MSCs proliferation and differentiation and ultimately accelerating bone formation, which involves various signaling pathways such as BMP/Smad, Wnt/β-catenin, and phosphatidylinositol 3-kinase (PI3K)/AKT [136].

#### 4.1.1. MSC-EVs Promote Osteogenic Differentiation by Enhancing the BMP/Smad Signaling Pathway

BMPs, members of the TGF-β superfamily, can promote bone formation by enhancing the osteogenic differentiation of BMSCs [137]. The BMP/Smad signaling pathway is mainly composed of BMPs, BMP receptors, Smad proteins, and other related transcription factors [138,139]. BMPs bind to serine/threonine kinase receptors and transduce signals through Smad-dependent and independent mechanisms [140,141]. Members of the BMP family bind to two distinct type II and type I serine/threonine kinase receptors, both of which are required for signal transduction [142]. BMPs bind to three distinct type II receptors, i.e., BMPR-II, ActR-II, and ActR-IIB. Regarding type I receptors, BMPs bind to three distinct type I receptors, including BMPR-IA, BMPR-IB, and ACVR-I [143,144]. Canonical BMP signaling is initiated when BMPs associate with type I and type II BMP receptors and form a multimeric receptor-ligand complex [145]. Within this complex, the constitutively phosphorylated type II BMP receptors activate type I BMP receptors by phosphorylation, initiating the canonical BMP signaling cascade [146]. Activated type I BMP receptors propagate BMP signals by phosphorylating Smad1/5/8. These Smads complex with Smad4, travel to the nucleus and act as activators and repressors of transcription of target genes, such as Runx2 and OSX, thus regulating osteogenic differentiation [147]. The activation of the BMP/Smad signaling pathway plays an important role in regulating the osteogenic differentiation of MSCs [148]. Recently, numerous studies showed that MSC-EVs promote osteogenic differentiation through the activation of the BMP/Smad signaling pathway, which may be ascribed to multiple MSC-EV-miRNAs [103,110]. Liu et al. [41] found that BMSC-OI-Exos exhibited 8 down-regulated miRNAs, including miR-877, and 16 up-regulated miRNAs (such as miR-328a-5p, miR-31a-5p, let-7a-5p, and let-7c-5p) during the process of promoting osteogenic differentiation in a rat calvarial defect model. Down-regulated miR-877 can up-regulate the expression of target genes involved in the BMP/Smad signaling pathway, such as BMPR-IA, BMPR-IB, and Smad1. Up-regulated miR-328a-5p, miR-31a-5p, let-7a-5p, and let-7c-5p can promote the down-regulation of targeted genes ACVR-I and ACVR-IIB, which inhibit the BMP/Smad signaling pathway [149]. Bioinformatics analysis showed that ACVR-IIB might be an antagonist of BMPR-II, and they are involved in a competitive and synergistic receptor activation network instead of a simple competition relationship between BMPR-II and ACVR-IIB [41]. BMSC-Exo-miRNAs mainly targeted ACVR-IIB/ACVR-I and regulated the competitive balance of BMPR-II/ACVR-IIB toward BMPR-elicited Smad1/5/9 phosphorylation, thus promoting osteogenic differentiation [41]. EVs obtained from different conditions may regulate BMP/Smad signaling pathway through different pathways. Huang et al. [110] generated functionally engineered Evs with enhanced osteoinductive function in vitro and in vivo, which were derived from genetically modified hBMSCs by constitutively expressing BMP-2. A study showed that 5 miRNAs targeting SMAD7 and SMURF1 were demonstrably upregulated in the BMP-2 hBMSC-Evs compared with hBMSC-Evs in vitro. SMURF1 and SMAD7 are well-characterized inhibitors of the BMP signaling pathway [150]. Further studies showed increased expression of miR-424 in the Evs. It has been proved that miR-424 played a role in down-regulating SMURF1 and potentiating BMP-2 signaling [151,152]. Above all, EV-miR-424 may potentiate BMP-2 signaling through down-regulating SMURF1, thereby enhancing the osteoinductive function of Evs. In addition, another study demonstrated that exosome mimetics (Ems) obtained from genetically modified hBMSCs in which expression of noggin, a natural BMP antagonist, was down-regulated could enhance the osteogenic properties of Ems. Further investigation suggested that treatment with Ems from noggin-suppressed hBMSCs with an osteogenic medium significantly decreased the expression of miR-29a, while the expression of BMPR-IA, phosphorylated Smad 1/5/8, and ID1 (a key downstream target of BMP signaling) was detected to significantly increase in MSCs compared to the control group [103]. Previous studies have shown that the knockdown of noggin in MSCs triggers key mediators of BMP signaling (Smad 1/5/8) and osteogenic molecules (Runx2 and OCN) [153]. The above results indicated that the enhanced osteogenic properties of EMs in which noggin was suppressed were mediated by inhibiting the expression of miR-29a to stimulate BMP/Smad signaling [103].

In summary, numerous studies have proved that BMP/Smad is a crucial signaling pathway in the osteogenic differentiation of MSCs. Engineered MSC-EVs can potentiate BMP signaling and promote Smad1/5/8 phosphorylation by altering the expression profile of miRNAs within them. Then, the key downstream targets of BMP signaling are up-regulated, including osteogenic molecules such as Runx2, accelerating osteogenic differentiation of MSCs. The regulation of the BMP/Smad signaling pathway involves the cumulative effect of multiple MSC-EV-miRNAs, such as miR-877 and miR-424 (as shown in Figure 3). However, current investigation regarding the miRNAs related to the positive or negative regulation of the BMP/Smad signaling pathway is still insufficient. In addition, DNA, proteins, lipids, and other active molecules in EVs may also be involved in the regulation of BMP/Smad signaling. Therefore, further studies are expected to explore the key bioactive molecules in EVs that can effectively regulate BMP/Smad signaling to better apply EVs in BTE.

#### 4.1.2. MSC-EVs Promote Osteogenic Differentiation by Regulating the Wnt/β-catenin Signaling Pathway

Coupled bone formation and bone resorption are a guarantee of bone homeostasis. Cumulative evidence demonstrates that Wnt signaling plays a pivotal role in regulating bone homeostasis, and it can be divided into canonical and noncanonical Wnt pathways according to whether β-catenin is involved [154]. Wnts can bind to cell surface receptor LRP5/6 and receptors encoded by the Frizzled gene family. Receptor activation, in turn, somehow activates Dishevelled (Dsh), the most proximal cytosolic component known. In canonical Wnt signaling, Dsh is involved in inhibiting the activation of glycogen synthesis kinase 3 (GSK-3), which is dedicated to the degradation of β-catenin. Inactivation of the GSK-3 allows β-catenin to accumulate within responding cells and to enter the nucleus, where it binds to T cell factor, the transcriptional effector of the Wnt pathway. This complex recruits other proteins to drive transcriptional activation of Wnt target genes [155]. Heo et al. [156] have demonstrated that the Wnt/β-catenin signaling pathway promotes osteogenic differentiation of PDLSCs by activating osteogenic transcription factors such as Runx2. However, recent studies indicate that the Wnt signaling pathway has an opposite dual effect during the process of promoting osteogenic differentiation of MSCs [157]. It was reported that the osteogenic differentiation of MSCs was partially inhibited at a concentration of 100 ng/mL of Wnt3a, whereas treatment with Wnt3a at both 5 and 25 ng/mL resulted in a significant relative increase in osteogenic differentiation, as manifested by increased AP activity and mineralization [158]. Liu et al. [159] demonstrated that the canonical signaling pathway was activated in inflammatory environments while noncanonical Wnt signaling was inhibited, which stimulates PDLSC differentiation and inhibits the terminal differentiation of mature osteoblasts. Therefore, canonical Wnt signaling appears to suppress osteogenic differentiation [160,161]. Further research showed that the expression of Wnt1, Wnt3a, and Wnt10a in inflammatory PDLSCs were significantly decreased after treatment with human PDLSC-derived Exos (hPDLSC-Exos), which contributes to the decrease in β-catenin and the increase in p-GSK3β. p-GSK3β is the most important negative regulator of the Wnt signaling pathway [162]. A study suggested that Exos can rescue the osteogenesis capacity of endogenous stem cells under an inflammatory environment and promote regeneration of alveolar bone by inhibiting the activation of the Wnt/β-catenin pathway [162]. Moreover, the optimal application concentration of hPDLSC-Exos is within a specific threshold [162]. Although numerous studies have shown that the negative effect of canonical Wnt signaling for the osteogenic differentiation of MSCs, there are still studies proving that the BMSC-Exos carrying mutant HIF-1α can activate the Wnt/β-catenin signaling pathway to promote new bone formation [111]. Therefore, the mechanism and effect of the Wnt/β-catenin signaling pathway regulated by MSC-EVs for osteogenic differentiation of MSCs need to be further discussed.

In summary, MSC-Evs can regulate the osteogenic differentiation of MSCs through the Wnt signaling pathway, which has an opposite dual effect on this process. Inhibition or overactivation of Wnt/β-catenin signaling perturbs the osteogenic differentiation of MSCs. The concentrations and activity of Wnt play a significant role in the dual effect of the Wnt signaling pathway. The overactivation of the canonical Wnt signaling at the site of damaged tissue can inhibit osteogenic differentiation of MSCs, which may be involved with the inflammatory environment. Engineered MSC-Evs can promote osteogenesis differentiation by inhibiting the activation of the Wnt/β-catenin signaling pathway. However, it has been shown that MSC-Evs can enhance osteogenesis by activating the canonical Wnt signaling in some studies. In addition, the inhibition of canonical Wnt signaling in the inflammatory environment by MSC-Evs also suggested that Evs may participate in immune regulation via this pathway. However, many issues, such as the optimal concentrations of MSC-Exos to reduce the overactivation of the Wnt/β-catenin pathway and enhance osteogenic differentiation of MSCs, are still not clear. Additionally, the duality of the Wnt signaling pathway in the regulation of osteogenic differentiation still needs to be further explored.

#### 4.1.3. MSC-Evs Promote Osteogenic Differentiation by Activating the PI3K/AKT Signaling Pathway

The PI3K/AKT signaling pathway plays a key role during the process of osteogenic and adipogenic differentiation of BMSCs [163]. As a lipid kinase located in the cytoplasm, PI3K is activated through the stimulation of upstream growth factors, and then PI3K transduces upstream signals from receptors to generate PIP3 [164]. Serving as a critical lipid second messenger, PIP3 recruits protein kinase AKT to the plasma membrane to promote AKT activation and then regulates osteogenic genes [165,166]. In the application of hiPS-MSC-Exos combined with β-TCP scaffolds to promote the repair of rat cranial defects, Zhang et al. [22] found that hiPS-MSC-Exos treatment resulted in a prominent increase in the positive effector genes expression of the PI3K/AKT pathway, such as PDGFA, FGF1/2, FGFR1, COL1A1/2, and BCL2L1. All these genes can positively modulate cell survival, proliferation, migration, osteogenic differentiation, or apoptotic inhibition [167,168,169,170]. The typical negative modulators of PI3K/AKT signaling, including GSK3β and PTEN, were remarkably decreased [167,168,169,170]. The results indicated that the PI3K/AKT signaling in hBMSCs is probably activated after the internalization of Exos, thus stimulating the proliferation and differentiation of hBMSCs. In addition, further investigation showed that the effect of hBMSC-Exos to enhance osteogenic differentiation may be CD73-mediated adenosine receptor activation of pro-survival AKT and MAPK/ERK signaling [136]. ATP is a danger signal released by cells in stress or injury to remove or destroy ‘‘dangers”, which will be subsequently degraded by ectonucleotidases, such as CD39, to ADP and AMP in a short time and finally degraded by CD73 into adenosine, a potent activator of the survival kinases, AKT and ERK [171]. Chew et al. [136] observed that adenosine signaling through phosphorylation of AKT and ERK played a significant role in mediating Exo-induced PDLSC migration and proliferation in a rat periodontal defect model. Compared to ERK, the attenuation of AKT phosphorylation leads to a greater reduction in PDLSC migration and proliferation. AKT, therefore, appears to be a more critical target of MSC-Exos to promote osteogenic differentiation.

In summary, MSC-EVs can promote osteogenic differentiation of MSCs by activating PI3K/AKT and MAPK/ERK signaling pathways after the internalization of EVs. Further research suggests that the enhanced cellular infiltration and proliferation observed during Exo-mediated periodontal regeneration could be attributed at least in part to exosomal CD73-mediated adenosine receptor activation of pro-survival AKT and ERK signaling. AKT is a key mediator in the above process. Moreover, the mediating process may be involved with the cell response to stress or injury. However, the contribution of stress-related mediating mechanisms to the activation of AKT and ERK pathways remains unclear. MSC-EVs cargo is highly diverse and complex and contains growth factors such as IGF and FGF that could potentially activate AKT and/or ERK signaling pathways in PDL cells. The other potential mediating pathways besides CD73-mediated adenosine receptor activation of pro-survival AKT and ERK signaling as well as require further exploration.

### 4.2. MSC-EVs Repair Bone Defects by Promoting Angiogenesis

Bone is a richly vascularized tissue. The vasculature, as a major transport pathway delivering hormones and growth factors to the bone, provides the bone with oxygen and nutrients and scavenges metabolites, which are critical for bone development and regeneration [172,173]. Furthermore, the blood supply can accelerate bone defect repair by promoting osteoblast migration and bone matrix mineralization, which are critical for bone regeneration in BTE. It has been reported that the major challenge after graft implantation is the maintenance of cell viability in the graft core, which depends on rapid angiogenesis within the tissue-engineered implants [173]. Studies have shown that MSC-EVs play a role in stimulating angiogenesis in bone, which is crucial to accelerating bone regeneration [174]. Researchers proved that BMSC-Exos preconditioned with a low dose of DMOG (DMOG-MSC-Exos) have superior pro-angiogenic properties and can promote bone regeneration in a critical-sized calvarial defect rat model, which may be related to the AKT/mTOR signaling pathway [106]. Compared with the control group, the expression of key components involved with angiogenesis in human umbilical vein endothelial cells (HUVECs) were significantly increased in the DMOG-MSC-Exos group. The molecules that regulated apoptosis were significantly down-regulated such as PTEN, p53, and p21 [106]. PTEN is an endogenous negative regulator of the PI3 kinase signal pathway. On the one hand, PTEN induces the inhibition of AKT kinase and mTOR kinase and inhibits the proliferation of endothelial progenitor cells [175,176]. On the other hand, when the AKT/mTOR pathway is activated, mTOR activation can lead to the down-regulation of PTEN, thus further alleviating its inhibition effect on the AKT/mTOR pathway, which creates a positive feedback loop to facilitate the activation of the pathway [177]. Further analysis showed that the superior pro-angiogenesis ability of DMOG-MSC-Exos was absent after the AKT/mTOR pathway was blocked with the AKT kinase inhibitor MK2206. All results proved that MSC-EVs promoting the angiogenesis of bone defect sites are involved in targeting the AKT/mTOR signaling pathway [106]. In addition, Wu et al. [53] showed that SHED-Exos can up-regulate the expression of angiogenesis-related genes (VEGF-A, KDR, FGF2, and SDF-1), thereby contributing to angiogenesis and osteogenesis in a rat alveolar bone defect model. The study found that adenosine monophosphate-activated protein kinase (AMPK) phosphorylation was increased in both HUVECs and BMSCs after stimulation with SHED-Exos. Additionally, the treatment of compound C, a pharmacological blocker of AMPK, partially reversed the impacts of SHED-Exos on the expression of angiogenesis-related genes (KDR, SDF-1) [53]. The above results suggest that AMPK signaling is activated by SHED-Exos to enhance angiogenesis. AMPK has been shown to exert both positive and negative effects on angiogenesis. The activation of AMPK signaling could attenuate angiogenesis induced by AKT/mTOR and other factors [178]. The specific mechanism of enhancing angiogenesis for EVs via AMPK signaling needs to be further investigated.

In summary, it is demonstrated that MSC-EVs can effectively promote bone regeneration by stimulating angiogenesis in the bone defect area, which may be closely related to the AKT/mTOR and AMPK signaling pathways. Although angiogenesis is critical in bone regeneration, the underlying mechanisms of MSC-EVs activating angiogenesis are not fully understood, and further research is needed. In addition, there are still many problems to be solved in the application of MSC-EVs in BTE: (1) What is the optimal dose of DMOG to treat MSCs for better enhancing angiogenesis of MSC-EVs? (2) What are the specific mechanisms of DMOG and other factors that affect the expression profile of EV cargo? (3) What is the most low-cost way to improve the pro-angiogenic capacity of EVs?

### 4.3. MSC-EVs Repair Bone Defects by Participating in Immune Regulation

Bone defects caused by trauma and tumors are usually characterized by peripheral inflammation and immune imbalance, which may also lead to implant failure of materials widely used in BTE [179,180]. Previous studies have shown that the immune microenvironment created by immune cells and their metabolites are the critical regulators of the microenvironment for bone regeneration, which has a significant influence on the activity of osteoblasts and osteoclasts and the regulation of bone deposition and strength by regulating the expression of chemokines, growth factors, and inflammatory factors [181,182]. Inflammatory stimulation recruits MSCs that receive pro-inflammatory signals (such as IL-6, IL-17, and TNF-α) to the injured site to improve the local microenvironment, which is conducive to tissue healing [183]. This may be attributed to the anti-inflammatory factors or EVs released by stimulated MSCs to mediate immunoreaction and regulate macrophage polarization [184]. Macrophages are important components of the innate immune system and play an important role in clearing pathogens and regulating inflammatory responses [185,186]. M1 macrophages mainly play a role in pro-inflammatory and immune clearance by secreting pro-inflammatory factors, thereby killing pathogens and tumor cells. However, excessive activation of M1 macrophages causes immune damage [187,188]. M2 macrophages are mainly involved in anti-inflammatory and wound healing by secreting anti-inflammatory and growth factors. Promoting phenotypic polarization of M2 macrophages can effectively promote angiogenesis and bone healing [187,188] (as shown in Figure 3). Numerous studies showed that MSC-EVs can not only promote the proliferation and differentiation of MSCs but also affect the activity of immune cells in a complex internal environment. It is believed that MSC-EVs could stimulate macrophages and other non-stem-like cells to secrete osteogenic growth factors such as BMP-2 and accelerate the healing of injured tissue [189,190]. In addition, it has been widely recognized that osteogenic differentiation can be enhanced by reducing inflammation [191]. Wang et al. [97] constructed a PCL composite scaffold with immunomodulatory potential by combining S-Nitrosoglutathione (GSNO) and MSC-Exos with a PCL scaffold that was surface modified by Poly(dopamine) coating. The results showed that, compared with the control group, Exos combined with GSNO significantly reduced the expression of inflammatory genes via internalization into macrophages. Due to the immunomodulating effects of GSNO by continually releasing nitric oxide (NO), this may be attributed to the regulatory effect of cargo within MSC-Exos or the synergistic effect of both GSNO and MSC-Exos [192]. To further prove whether the bioactive molecules of MSC-Exos play a key role in the process of regulating the immune response, Li et al. [62] analyzed the role of highly expressed miR-451a in ADSC-Exos during the repair of rat cranial defects. Through transfecting miR-451a analogues and inhibitors into macrophages, it was found that miR-451a can directly regulate the expression of macrophage migration inhibitory factor (MIF) by specifically binding to the 3′UTR of MIF mRNA, thereby promoting M1-to-M2 polarization of macrophages [62]. MIF is a multifactorial pro-inflammatory mediator involved in immune regulation [193]. Therefore, the above results demonstrated that the miR-451a derived from MSC-Exos is an effective anti-inflammatory factor. Moreover, it also confirmed that bioactive molecules within MSC-Exos play a vital role in the process of regulating immune response [97,194]. Previous studies have also shown that miRNAs, such as let-7b, or proteins within MSC-Exos pretreated by lipopolysaccharide can reduce inflammation [195]. Further studies demonstrated that MSC-EVs may promote bone regeneration by activating the OPG/RANK/RANKL signaling pathways [190,196]. The bone immunoregulatory pathway OPG/RANK/RANKL participates in bone resorption by regulating osteoclast activity and inflammatory response [197]. RANKL secreted by osteoblasts can activate osteoclasts by binding to RANK [198]. OPG can competitively bind to RANKL to block the RANKL-RANK binding, thereby inhibiting osteoclast activity [199]. Therefore, the balance of bone metabolism depends on the expression ratio of OPG to RANKL (as shown in Figure 3). Shi et al. [195] proved that BMSC-EVs inhibited osteoclast activities by regulating the expression of OPG and RANKL in the therapy of periodontitis rats by combining hydrogels with BMSC-EVs.

In summary, MSC-EVs promote M1-to-M2 polarization of macrophages by delivering their internal cargo, thereby participating in immune regulation in bone defect repair. In addition, MSC-EVs can inhibit osteoclast activity by regulating the OPG/RANK/RANKL signaling pathway, thereby inhibiting bone resorption. Therefore, MSC-EVs may promote bone repair by reducing the inflammatory response and inhibiting the activity of osteoclasts.

## 5. Future Perspectives

In MSC-EV-mediated BTE, bone graft materials such as scaffolds and hydrogels are often used alone or combined as the bridging structure to repair bone defects. By selecting suitable donor and parent cells, EVs with good osteoinductive potential could be obtained, and the EVs will be loaded onto the ideal bone graft materials to promote bone tissue regeneration at the defect sites. In this paper, we comprehensively reviewed the current status of bioactive materials combined with MSC-EVs utilized in repairing bone defects and the mechanism of MSC-EVs promoting bone regeneration in BTE. Firstly, the combination of hydrogels and scaffolds has achieved some success in terms of the application of MSC-EVs in bone defects. Due to the poor mechanical properties of hydrogels and the low osteoinductive activity and inefficient filling of irregular gaps of scaffolds, the combination of hydrogels and scaffolds is often used to effectively deliver EVs, which is the most common strategy used in BTE currently. Secondly, based on the combined application of hydrogel and scaffold materials, a variety of innovative ways to enhance the delivery efficiency of MSC-EVs have been developed. Owing to most current carriers explosively releasing EVs, it leads to inadequate therapeutic effects and even causes bone resorption. Researchers achieve efficient transport, sustained release, and long-term storage of EVs by developing innovative scaffolds and modifying composite hydrogel scaffolds or combining bone graft materials with tethering peptides and other active substances. Lastly, enhancing the osteoinductive capacity of EVs in various ways is the key factor in promoting bone defect repair in BTE. The limited osteoinduction capacity of EVs is not sufficient to effectively promote bone formation in critical bone defects when used EVs alone. Therefore, EVs are often modified by gene modification, hypoxic environment culture, chemical reagent pretreatment, or combined with other active substances to prepare EVs with enhanced osteoinductive ability. In addition to the specific application of MSC-EVs combined with bioactive materials, numerous studies explored the mechanism of EVs promoting bone defect repair in BTE. Studies found that engineered MSC-EVs may regulate the key signaling pathways of osteogenic differentiation, including BMP/Smad, Wnt/β-catenin, and PI3K/AKT signaling pathways, by altering the levels of miRNAs and other active substances within them. Enhancing the angiogenesis mechanism of MSC-EVs is involved with the activation of AKT/mTOR and AMPK signaling pathways. Moreover, many studies have shown that EVs can participate in immune regulation and reduce inflammatory response by promoting M1-to-M2 polarization of macrophages. Further research proved that it may be associated with the inhibition of osteoclast activities by regulating the OPG/RANK/RANKL signaling pathway, then inhibiting bone resorption and accelerating injury repair (as shown in Figure 2 and Figure 3).

EV-based cell-free bone regeneration strategy can avoid the disadvantages of MSC transplantation, such as ethical issues, immune rejection, and tumor risk, showing infinite application potential. However, it is undeniable that the research on the effect of EVs on bone regeneration is still in the preliminary stage. There are still many issues to be resolved conveniently and reliably, such as the ineffective delivery of EVs, the long-term preservation of EVs, the uncertainty of the optimal treatment dose of EVs, and the unstandardized porosity value or pore size of a biocompatible 3D porous scaffold for bone defects. Therefore, it is necessary to further explore and clarify the key factors to improve the therapeutic effect of EVs in the future. For instance, in the delivery of EVs, future studies need to determine the proportion of components in the composite bone graft materials. Further optimizing the mechanical properties and biological activity of bone graft materials to obtain widely recognized grafts is significant in BTE. In terms of the performance improvement of EVs, as the modified MSC-EVs may produce some unexpected therapeutic effects, it is necessary to carefully evaluate whether there are the disadvantages of high cost, poor pharmacokinetics, low efficiency, and potential side effects. In addition, studies have shown that MSC-EVs are rich in RNAs, proteins, and lipids. Theoretically, EVs are likely to regulate bone formation through the interaction of their various components. However, most studies are limited to the role of miRNAs encapsulated in EVs in BTE, suggesting that the mechanism of EVs in bone regeneration is far from being explored. Therefore, the effective active molecules of different types of MSC-EVs promoting osteogenic differentiation, angiogenesis, inflammatory regulation, and osteoclast inhibition need to be further clarified.

## 6. Discussion and Conclusions

The strength of this review is that it comprehensively summarized the latest progress in applying MSC-EVs in BTE to promote bone defect repair and the potential mechanism of MSC-EVs promoting bone regeneration, which is of great significance for the currently hot research on bone defect repair. This review systematically and comprehensively describes the latest application progress of MSC-EVs in bone defect repair by summarizing the latest studies, for example, the development and improvement of new hydrogel and scaffold materials and their advantages and disadvantages currently; recent advances and potential risks of EVs delivery by the combination of hydrogel and scaffold materials; and effective advances in improving EVs performance by gene engineering or other methods. In addition, in order to make readers better understand the significance of MSC-EVs in bone defect repair, the potential mechanism of MSC-EVs in BTE was systematically described in detail from the aspects of promoting osteogenesis and angiogenesis, inhibiting osteoclast formation and participating in immune regulation. This is also one of the highlights of this paper, which is of great significance for the application of MSC-EVs in basic research and clinical treatment to promote bone regeneration. In a word, our review provides the latest evidence and views on the effective application, potential risks and solutions of MSC-EVs combined with hydrogels and/or scaffolds and the potential mechanism of MSC-EVs enhancing bone regeneration. However, this paper may also have many limitations. Firstly, this paper focuses on the characteristics and application of MSC-EVs combined with hydrogel and/or scaffold materials for repairing bone defects, while there is little description of the preparation process of bone graft materials in detail, which may hinder the corresponding animal research to some extent. Secondly, this paper mainly compares several common MSC-EVs due to the wide variety of MSCs; therefore, the evidence for the application of EVs parental cell selection may not be sufficient. Thirdly, owing to the limited space of this paper, it is not possible to list and elaborate on all the independent, innovative hydrogel scaffold composites with different characteristics. Therefore, in the future, it is necessary not only to continue to explore the effective application of EVs combined with bioactive carrier materials but also to further clarify the active substances in MSC-EVs and their potential mechanisms for promoting bone regeneration.

In conclusion, although preclinical studies have reported promising results, several enhancements of MSC-EV therapy are required to obtain superior outcomes. Further research on effective delivery, activity maintenance, controlled, sustained release and potential mechanisms of EVs will become key factors for the safe and effective application of EVs in bone defect repair.

## Figures and Tables

**Figure 1 membranes-12-00716-f001:**
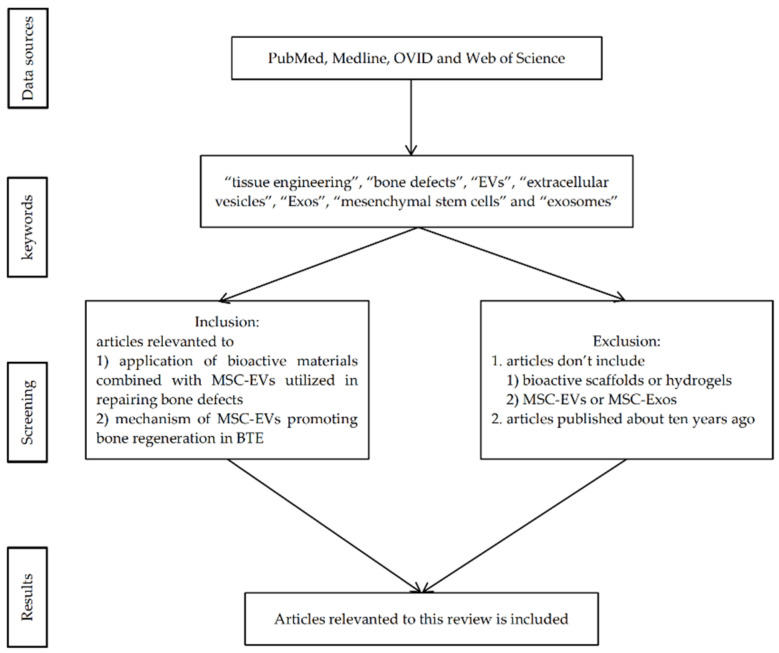
The flowchart for literature screening.

**Figure 2 membranes-12-00716-f002:**
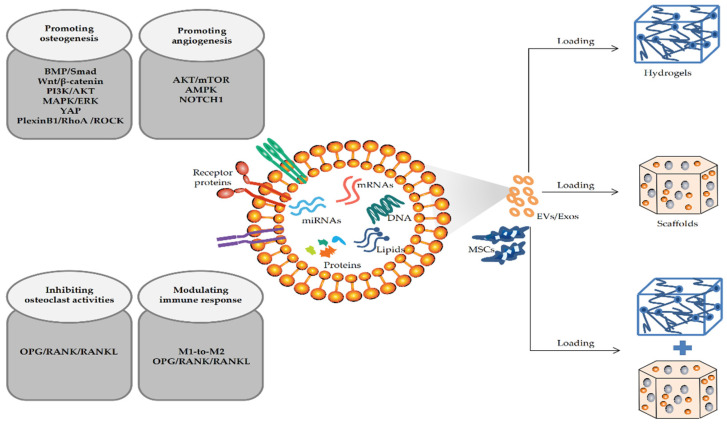
Mechanisms of MSC-EVs/MSC-Exos promoting the repair of bone defects in BTE.

**Figure 3 membranes-12-00716-f003:**
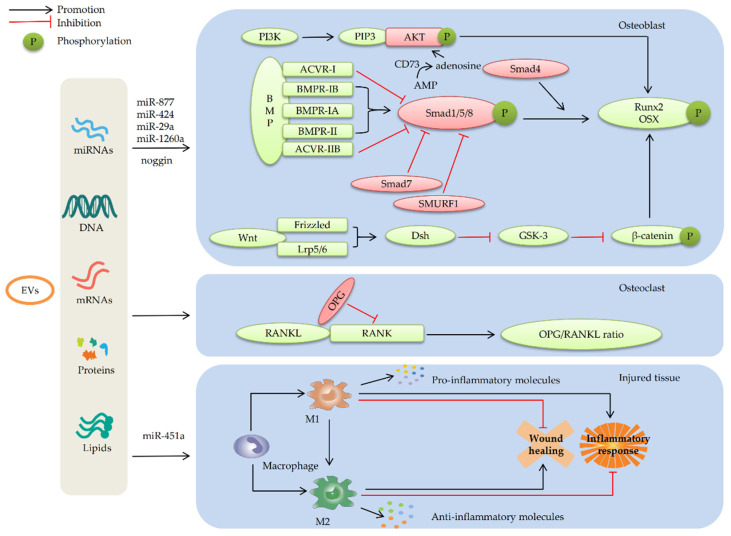
The main mechanism of EVs to improve the activity of osteoblasts, inhibit the activity of osteoclasts and reduce inflammation through the intrinsic cargo within EVs.

**Table 1 membranes-12-00716-t001:** Comparison of application characteristics and osteogenesis mechanism of common EVs parent cells.

Parent Cells	Application Characteristics	Functions	References
BMSCs	Effective osteogenesis Easy to obtain Most widely used	Osteogenesis differentiation	[37,38]
hADSCs	Easy to obtain Rapid multiplication; the best choice to increase yield The most widely distributed in humans Poor osteogenic ability and require additional substances to induce osteogenesis	Osteogenesis differentiation	[39,40,41,42]
hUCMSCs	Higher pluripotency Strongest angiogenic properties Source from the waste organization, abundant sources There are no ethical and moral disputes Higher clinical potential	Angiogenesis	[43,44,45,46,47]
iPS-MSCs	There are advantages both of iPSCs and MSCs Unlimited growth and self-renewal No longer tumorigenic There are no ethical and moral disputes Stronger proliferation capacity and immune regulation function	Osteogenic differentiation and angiogenesis	[15,22,48,49]
SHEDs	Multiple differentiation potential Non-invasive means to obtain (easy access) There are no ethical issues Rich in growth factors such as FGF2, TGF-β2 Stronger proliferative capacity	Osteogenic differentiation and angiogenesis	[50,51,52,53]

**Table 2 membranes-12-00716-t002:** Summary of the application of different carrier materials in BTE.

Materials	Advantages	Disadvantages	Common Types	Application	References
Hydrogels	Similar to the 3D environment in vivo Effectively encapsulate EVs to maintain local concentrations and enhance EVs performance Effectively fill irregular defect environment Release EVs slowly and sustainably Targeted transport, reducing loss and ectopic effect Good biocompatibility and chemical activity	Poor mechanical properties Poor stability Inadequate adhesion of cell Failure of long-term retained of EVs	Natural materials, (Gelatin; HA-Gel; chitosan) Synthetic polymers, (PEG)High-performance composite hydrogels, (modified injectable thermosensitive hydrogels; composite hydrogels with enhanced mechanical properties)	Enhancing the performance of hydrogels(modifying hydrogels; combination application of different hydrogels) Improving the transport efficiency of EVs (adding fixed peptides; construction of fusion polypeptides)	[29,57,58,59,60,61,62,63,64,65,66,67,72]
Scaffolds	The 3D pore structure is similar to natural bone and provides space for the growth and vascularization of new tissue Good mechanical properties Absorbable and biodegradable Specific inducible surface stimuli enhance the activity of EVs	Failure of EVs Slow releasing Risk of missing the target Unable to provide similar living environments in vivo Poor effect of filling irregular voids	Classical scaffold materials (collagen sponge, bone cement scaffold, BG; β-TCP, HA scaffolds; polymer scaffolds) Innovative synthetic scaffolds	Enhancing the activity of EVs (preconditioning MSCs;inducing the expression of osteogenic related genes or proteins; combined with small molecule drugs and inducible factors such as siRNAs (externally and externally loaded)) Realizing the slow and sustained release of EVs(innovative synthetic scaffolds; scaffold materials combined with other materials;scaffold materials that provides EVs lyophilization protection)	[19,22,77,78,88,89,90,92,98,102,103,104,105,106,111,113]
Hydrogels + Scaffolds	Effectively encapsulate EVs and enhance EVs activity Sustain and slow release of EVs Effective and efficient delivery of EVs Good effect of filling bone defects Stable mechanical properties Good biocompatibility Long-term retained of EVs	The synthesis of composite materials is complicated The quality of application varies	Hydrogels filling into scaffold materials (HA-Gel hydrogels combined with nHP scaffolds; PLGA-PEG-PLGA gel microspheres combined with PLLA scaffolds) Forming new composite materials (omposite material PG/TCP; Self-healing composites)	Various new composite materials with good mechanical properties, such as self-healing, stability, adhesion and antibacterial abilities, were obtained	[47,56,77,78,115,117,118,119]

## Data Availability

Not applicable.

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
