# Peer review of "Mesenchymal Stem Cell-Derived Extracellular Vesicles for Bone Defect Repair"

_membranes, 2022, doi:10.3390/membranes12070716_

Round 1

Reviewer 1 Report

The manuscript assessed “Mesenchymal Stem Cell-derived Extracellular Vesicles for Bone Defect Repair ”. This research is under the scope of this Journal.

However, there are some concerns about the present manuscript:

  •  

(Keywords)  

  • Please order the keywords / Mesh Terms alphabetically for a standardized presentation of the keywords.

(Introduction)

  • The “Critical size defect” need to be better supported in bone regeneration with CSD in an animal study. Please read these references. https://doi.org/10.3390/molecules26051339; https://doi.org/10.1080/08941939.2016.1241840.
  •  
  • What is the importance of this review study? Which results are comparable with other articles (https://doi.org/10.3390/ph14040289)? What has this study been new? 

(M&M)

  • This section would better communicate to readers if restructured. A flowchart or diagram of the article selection would be valuable.

(Results)

  • In Tables 1 and 2, add a column with references to each situation.  
  • Improve the resolution quality of all figures and graphs (and a presentation). The font/language in the figure/caption is different from the text. Please, standardise the size and the font in the figures with the font of the manuscript. 

(Discussion)

  • Can you discuss the importance MSC-EVs to repair bone defects in scaffolds of the pores or the space provision versus compacted materials, please read this article, Palma et al. (2010, New formulations for space provision and bone regeneration. Biodental Eng. I, 71-76) reported the influence of different formulations of bone grafts in providing an adequate scaffold, thus emphasizing the importance of the type of carrier in the three-dimensional distribution of particles and also space provision in new bone formation. 
  • Please, identified what was the strength(s) and limitations of this study? And also, implications for future perspectives before the conclusions.

(Conclusions)

  • The conclusion section should more thoroughly summarize the results, this section is too long and sometimes too ambiguous 

(References)

  • Check the reference MDPI format in the manuscript and the references. The references have a different format one the manuscript presentation.
  •  

Author Response

Dear Editors and Reviewers,

We are submitting our revised manuscript entitled “Mesenchymal Stem Cell-derived Extracellular Vesicles for Bone Defect Repair” (membranes-1817446) to Membranes for your reconsideration of its suitability for publication. All authors have read and approved the revised manuscript. We deeply appreciate the time and effort you have spent in reviewing our manuscript. We have learned much from the reviewers’ comments, which are valuable and constructive. After studying the comments and your advice carefully, we have made a corresponding revision. Words in red are the changes we have made in the new manuscript. Our responses to the comments are listed below:

Reviewer1

Major comments:

(Keywords)  

  • Please order the keywords / Mesh Terms alphabetically for a standardized presentation of the keywords.

Reply: Thanks for your suggestion. We have made corresponding changes in the text.

(Introduction)

  • The “Critical size defect” need to be better supported in bone regeneration with CSD in an animal study. Please read these references. https://doi.org/10.3390/molecules26051339; https://doi.org/10.1080/08941939.2016.1241840.

Reply: Thanks for your constructive suggestion. we have supplemented the description of “Critical size defect” and added corresponding literature as support in the introduction. Thanks again for your suggestion.

  • What is the importance of this review study? Which results are comparable with other articles (https://doi.org/10.3390/ph14040289)? What has this study been new? 

Reply: Thanks for your comments. The importance of this review is that it comprehensively summarized the latest progress in applying MSC-EVs in bone tissue engineering to promote bone defect repair and the potential mechanism of MSC-EVs promoting bone regeneration, which is of great significance for the currently hot research on bone defect repair. Accumulated studies have shown that EVs with osteogenic inductive properties can be derived from a variety of cells, including osteocytes, chondrocytes, and mesenchymal stem cells. However, most studies of cell-derived EVs except MSC-EVs have not shown the ideal therapeutic effect on the regeneration and repair of critical size bone defects. Therefore, existing evidence suggests that MSC-EVs are crucial in bone regeneration and repair. Compared with other reviews, this paper details the latest application progress of MSC-EVs, which are the most widely used in bone tissue engineering and have infinite potential therapeutic value, combined with hydrogels and/or scaffolds and other bioactive molecules in repairing bone defects. In addition, the therapeutic effects and application characteristics of different sources of MSC-EVs are summarized in this paper, which is of great significance for basic research and clinical application of MSC-EVs to promote osteogenesis or angiogenesis. It is worth mentioning that this manuscript not only expounds on the specific application of MSC-EVs combined with hydrogels and/or scaffolds in bone tissue engineering but also provides a more comprehensive summary of the potential mechanisms of MSC-EVs themselves or as drug carriers to promote bone regeneration, enabling readers to have a more comprehensive and in-depth understanding of the related content of MSC-EVs in promoting bone defect repair, which is also one of the highlights of the article. Overall, our review provides the latest evidence and perspectives on the effective application, potential risks and solutions of MSC-EVs combined with hydrogels and/or scaffolds and the potential mechanism of MSC-EVs enhancing bone regeneration.

We have modified and supplemented the purpose and significance of this paper in the part of the Introduction. Thank you very much for your suggestions and reminders, which are of great significance for us to improve the quality of this manuscript.

(M&M)

  • This section would better communicate to readers if restructured. A flowchart or diagram of the article selection would be valuable.

Reply: Thanks for your constructive suggestion. A flowchart or diagram of the article selection is indeed a better way to communicate to readers. We have added a flowchart of the article selection in this section. Thanks again for your suggestion.

(Results)

  • In Tables 1 and 2, add a column with references to each situation.  

Reply: Thanks for your suggestion, we have modified Tables 1 and 2 appropriately and added a column with references to each situation in the far-right column.

  • Improve the resolution quality of all figures and graphs (and a presentation). The font/language in the figure/caption is different from the text. Please, standardize the size and the font in the figures with the font of the manuscript. 

Reply: Thanks for your thoughtful suggestion, we have improved the resolution quality of all figures and standardized the size and the font in the figures with the font of the manuscript. In addition, in order to display the application of MSC-EVs combined with hydrogels and/or scaffolds more vividly, we have appropriately optimized Figure 2 in the article. Thank you again for your advice.

(Discussion)

  • Can you discuss the importance MSC-EVs to repair bone defects in scaffolds of the pores or the space provision versus compacted materials, please read this article, Palma et al. (2010, New formulations for space provision and bone regeneration. Biodental Eng. I, 71-76) reported the influence of different formulations of bone grafts in providing an adequate scaffold, thus emphasizing the importance of the type of carrier in the three-dimensional distribution of particles and also space provision in new bone formation. 

Reply: Thanks for your kind advice. We agree with your suggestion that the special 3D porous structure of scaffolds is necessary for the distribution of MSC-EVs and the promotion of subsequent bone regeneration, while the importance of the porous structure of scaffolds has been relatively little described in this paper. Therefore, after carefully reading the literature recommended by you, we consulted some relevant references and supplemented the relevant content in the corresponding part of our manuscript (Section3.2.2; Page 8, paragraph 1). Thank you again for your constructive suggestion.

The specific supplementary contents are as follows:

Palma et al. reported the influence of different formulations of bone grafts in providing an adequate scaffold, thus emphasizing the importance of the type of carrier in the three-dimensional distribution of particles and also space provision in new bone formation. The results showed that the lyophilized form carrier created a more homogenous interparticle spacing, allowed a more suitable particle distribution and stabilization and provided a required space which is crucial for proper cellular and vascular colonization, then promoting a faster bone regeneration with relevant clinical benefits. Similarly, compared with compacted materials, a biocompatible 3D porous scaffold could ensure a uniform spacing and stable distribution of MSC-EVs. The pores or the space provision of scaffolds can assure an adequate environment for growth factors and nutrients, realizing the constant flow of nutrients, cells, and growth factors from the outer portion to the core of the scaffold and promoting bone regeneration. In addition, scaffold surface porosity and the related micro and nano-morphology directly influence the cell behavior, stimulating proper communication among the resident cells. There is no consensus concerning the more appropriate porosity value or pore size currently. However, when mechanical properties are satisfied, over 90% of studies recommended high porosity values, with a wide range of pore sizes from 10 to at least 200 µm. (Section3.2.2; Page 8, paragraph 1)

  • Please, identified what was the strength(s) and limitations of this study? And also, implications for future perspectives before the conclusions.

Reply: Thanks for your kind suggestion. We have added a description of the strength and limitations of this study in the part of the discussion. And we have also supplemented and improved implications for future perspectives before the conclusions.

The details of the modification are as follows:

The strength of this review is that it comprehensively summarized the latest progress in applying MSC-EVs in bone tissue engineering to promote bone defect repair and the potential mechanism of MSC-EVs promoting bone regeneration, which is of great significance for the currently hot research on bone defect repair. Compared with other similar studies, our review systematically and comprehensively describes the latest application progress of MSC-EVs in bone defect repair by summarizing the latest studies. For example, the development and improvement of new hydrogel and scaffold materials and their advantages and disadvantages currently; recent advances and potential risks of EVs delivery by the combination of hydrogel and scaffold materials; effective advances in improving EVs performance by gene engineering or other methods. In addition, in order to make readers better understand the significance of MSC-EVs in bone defect repair, the potential mechanism of MSC-EVs in BTE was systematically described in detail from the aspects of promoting osteogenesis and angiogenesis, inhibiting osteoclast formation and participating in immune regulation. This is also one of the highlights of this paper, which is of great significance for the application of MSC-EVs in basic research and clinical treatment to promote bone regeneration. In a word, our review provides the latest evidence and views on the effective application, potential risks and solutions of MSC-EVs combined with hydrogels and/or scaffolds and the potential mechanism of MSC-EVs enhancing bone regeneration. However, this paper may also have some limitations. Firstly, this paper focuses on the characteristics and application of MSC-EVs combined with hydrogel and/or scaffold materials for repairing bone defects while there is little description of the preparation process of bone graft materials in detail, which may hinder the corresponding animal research to some extent. Secondly, this paper mainly compared several common MSC-EVs due to the wide variety of MSCs, therefore the evidence for the application of EVs parental cell selection may not be sufficient. Thirdly, owing to the limited space of this paper, it is not possible to list and elaborate on all the independent innovative hydrogel scaffold composites with different characteristics. Therefore, in the future, it is necessary not only to continue to explore the effective application of EVs combined with bioactive carrier materials but also to further clarify the active substances in MSC-EVs and their potential mechanism for promoting bone regeneration.

(Conclusions)

  • The conclusion section should more thoroughly summarize the results, this section is too long and sometimes too ambiguous 

Reply: Thank you very much for pointing out our problems, which is of great significance for us to find our shortcomings and improve our article. Through reading your suggestions carefully, we have appropriately deleted and modified the conclusion paragraph of this manuscript and added discussion and future perspectives sections to make the article clearer and more organized. Thank you again for your advice.

(References)

  • Check the reference MDPI format in the manuscript and the references. The references have a different format one the manuscript presentation.

Reply: Thank you very much for your kind suggestion. We have modified the formatting of the references in this manuscript according to the formatting requirements of the MDPI.

Thanks and Best regards!

Yours Sincerely,

Xuchang Zhou

07/17/2022

Reviewer 2 Report

Dear editor,

The manuscript (membranes-1817446) aims to review mesenchymal stem cell-derived extracellular vesicles coupled with different carrier systems for bone defect repair. The topic is interesting and the manuscript reads well. However, there are a few concerns that should be addressed to make the work re-considerable for publication:

1- A simple search in the literature implies that there are several similar/relevant studies such as: https://www.frontiersin.org/articles/10.3389/fbioe.2019.00352/full. I was wondering what novel perspective, information, etc, are presented in the current manuscript. The relevant studies should be cited and compared with the current work.

2- Page 4, paragraph 2; the summary paragraph (in general summary paragraphs) is unnecessary.

3- Figure, what scaffold means? Hydrogel can not act as a cell physically/biochemically supporting scaffold itself? If yes, the headlines should be changed taking into account the hydrogel scaffolds.

4- How new/updated are the references?

5- The parts related to signalling pathways exclude the likely effect of carrier material and its degradation byproducts. How different could be the performance of EVs alone and as combined with synthetic/natural scaffolding/carrier materials?

6- How much the performance of EV is influenced by the environmental conditions including temperature, pH, etc.? biodegradation of scaffold materials can potentially alter pH and indirectly affects EV.

Author Response

Dear Editors and Reviewers,

We are submitting our revised manuscript entitled “Mesenchymal Stem Cell-derived Extracellular Vesicles for Bone Defect Repair” (membranes-1817446) to Membranes for your reconsideration of its suitability for publication. All authors have read and approved the revised manuscript. We deeply appreciate the time and effort you have spent in reviewing our manuscript. We have learned much from the reviewers’ comments, which are valuable and constructive. After studying the comments and your advice carefully, we have made a corresponding revision. Words in red are the changes we have made in the new manuscript. Our responses to the comments are listed below:

Reviewer2

  • A simple search in the literature implies that there are several similar/relevant studies such as: https://www.frontiersin.org/articles/10.3389/fbioe.2019.00352/full. I was wondering what novel perspective, information, etc, are presented in the current manuscript. The relevant studies should be cited and compared with the current work.

Reply: Thank you very much for your kind suggestion. We have carefully read the relevant literature recommended by you, which is of great significance to the improvement of the content and quality of our manuscript. Compared with other similar studies, our review has a more systematic and comprehensive description of the latest application progress of MSC-EVs in bone defect repair by summarizing the latest studies, especially in BTE. For example, the development and improvement of new hydrogel and scaffold materials and their advantages and disadvantages currently; recent advances and potential risks of EVs delivery by the combination of hydrogel and scaffold materials; effective advances in improving EVs performance by gene engineering or other methods. In addition, in order to make the reader better understand the significance of MSC-EVs in bone defect repair, the potential mechanism of MSC-EVs in BTE was systematically described in detail from the aspects of promoting osteogenesis and angiogenesis, inhibiting osteoclast formation and participating in immune regulation. This is also one of the highlights of this paper, which is of great significance for the application of MSC-EVs in basic research and clinical treatment to promote bone regeneration. Thank you very much for your suggestions, and we have supplemented the content and added references in the introduction and discussion part (combined with the opinions of the two reviewers, we decided to add discussion and future perspectives part and modified the content of the conclusion part), hoping to improve the significance of this article for readers and meet your expectations. Thank you again for your kind suggestion.

2- Page 4, paragraph 2; the summary paragraph (in general summary paragraphs) is unnecessary.

Reply: Thank you very much for your kind advice. We fully agree with your suggestion after careful consideration and we have made corresponding modifications in the article, deleting Page 4, Paragraph 2.

3- Figure, what scaffold means? Hydrogel can not act as a cell physically/biochemically supporting scaffold itself? If yes, the headlines should be changed taking into account the hydrogel scaffolds.

Reply: Thank you very much for your careful reading and valuable suggestions. First, the scaffolds in this paper refer to non-gel solid scaffolds. Due to the poor mechanical properties, hydrogels generally are not directly used in bone defect repair which has a high requirement for mechanical performance. However, hydrogels can function as a transport vehicle for the controlled release of EVs/Exos, or as a filler material to fill solid scaffolds with larger pores for repairing bone defects. In addition, section 3.2.3 in this review mainly describes the most common application of the hydrogels combined with scaffolds, which form hydrogel composite materials or composite scaffold materials to deliver EVs/Exos. Therefore, we directly listed hydrogel as a material for EVs/Exos transport or a bone defect filling material rather than hydrogel scaffolds. Thank you again for your very instructive advice. We have benefited a lot.

4- How new/updated are the references?

Reply: Thanks for your comment. According to statistics, we found that references cited in this paper in recent five years (2017~2022) account for more than 50%, and references in recent ten years (2012~2022) account for more than 80%.

5- The parts related to signalling pathways exclude the likely effect of carrier material and its degradation byproducts. How different could be the performance of EVs alone and as combined with synthetic/natural scaffolding/carrier materials?

Reply: Thanks for your constructive suggestion, it is a very interesting and meaningful issue. In fact, the signalling pathways section of our article contains both literature you mentioned. All EVs-related literature in bone defect repair is included in this manuscript. However, most of the current studies are limited to using material carriers for EVs delivery, which could provide a better environment for EVs. For example, biological materials can control the release of EVs and prolong the half-life of EVs; or biological materials can create a good living environment for EVs to maintain their biological activities. Overall, current research is limited to exploring the functions of materials that enable efficient delivery of EVs and promote the maintenance of EV activity, without additionally conferring new functions on EVs by activating new signaling pathways. However, we agree with you very much. It is possible that the function of EVs may be strengthened due to the existence of materials, which will be a worthwhile topic to explore in the future.

6- How much the performance of EV is influenced by the environmental conditions including temperature, pH, etc.? biodegradation of scaffold materials can potentially alter pH and indirectly affects EV.

Reply: Thanks for your constructive suggestion very much, which is of great significance for us to further consider the potential therapy mechanism of hydrogels and/or scaffolds combined with EVs in bone defect repair. Studies on the effects of scaffold materials biodegradation on injured tissues or active molecules are most seen in the immune response: To achieve effective bone defect repair, the carrier material is required to have good biodegradability and biocompatibility, while the bioincompatible or inappropriate degradation of the scaffold materials may cause inflammation and affect the microenvironment of bone regeneration. we have mentioned the role of EVs in immune modulation in this paper (Section 4.3). Although there is no specific description, the role of EVs in immune modulation may be partly related to the effect of scaffold material biodegradation. After further searching for relevant literature, we found that most of the current literature focused on the study of the degradation characteristics of scaffolds. For example, pH value is lower in inflammatory conditions, and pH-responsive smart hydrogel can offer targeted controlled release behavior to wound area; Photo-responsive hydrogel goes through light-mediated degradation. We have described the biodegradation of scaffolds in this paper, but no article mentions that the biodegradation of scaffolds can change the environmental conditions such as pH, temperature, etc., thereby changing the therapeutic effect of EVs in bone regeneration. In our opinion, we agree with you very much that the biodegradation of scaffold materials may potentially alter environmental conditions and indirectly affect the release and effect of EVs. However, no relevant descriptions are being found for this issue, which requires further research and exploration. Thank you again for your very valuable advice.

We hope the Reviewers will be satisfied with the revisions for our manuscript. If you have any questions about our review, please do not hesitate to contact us.

Thanks and Best regards!

Yours Sincerely,

Xuchang Zhou

07/17/2022

Round 2

Reviewer 1 Report

The authors improve the manuscript, following the reviewer's indications. Congratulations!

Note: The reference number 79 is incomplete, please add the complete reference below.

(79. Palma, P. J., Matos, S., Ramos, J., Guerra, F., Figueiredo, M. H., & Krauser, J. (2010). New formulations for space provision and bone regeneration. Biodental Eng. I1, 71-76. WOS:000282776500012; SBN 978-0-415-57394-8.)

Reviewer 2 Report

Dear editor,

Considering the proper revision of the manuscript, I recommend it for publication.